# Discriminative Dissolution Method Using the Open-Loop Configuration of the USP IV Apparatus to Compare Dissolution Profiles of Metoprolol Tartrate Immediate-Release Tablets: Use of Kinetic Parameters

**DOI:** 10.3390/pharmaceutics15092191

**Published:** 2023-08-24

**Authors:** Bruno Solis-Cruz, Daniel Hernandez-Patlan, Elvia A. Morales Hipólito, Guillermo Tellez-Isaias, Alejandro Alcántara Pineda, Raquel López-Arellano

**Affiliations:** 1Laboratory 5: LEDEFAR, Multidisciplinary Research Unit, Superior Studies Faculty at Cuautitlan (FESC), National Autonomous University of Mexico (UNAM), Cuautitlan Izcalli 54714, Mexico; bruno_sc@comunidad.unam.mx (B.S.-C.); eadriana_mh@yahoo.com.mx (E.A.M.H.); 2Nanotechnology Engineering Division, Polytechnic University of the Valley of Mexico, Tultitlan 54910, Mexico; 3Division of Agriculture, Department of Poultry Science, University of Arkansas, Fayetteville, AR 72701, USA; gtellez@uark.edu; 4LUAL Asesores, Iztapalapa, Mexico City 09438, Mexico; alejandro.alcantara.pineda@gmail.com

**Keywords:** USP IV apparatus, open-loop configuration, dissolution profiles, kinetic parameters, generic drugs, similarity, dependent methods of comparison, independent methods of comparison

## Abstract

The use of the USP IV apparatus (flow-through cell) has gained acceptance in recent years due to its versatility and ability to discriminate due to its hydrodynamic conditions. Therefore, the objective of the present study was to develop a discriminative dissolution method in the USP IV apparatus using the open-loop configuration, as well as to propose a method to compare non-cumulative dissolution profiles obtained in the open-loop configuration considering kinetic parameters and validate its predictive power through its comparison with independent and dependent methods using five commercial immediate-release tablet drugs (one reference drug and four generic drugs) of metoprolol tartrate as a model drug. The comparison of the non-accumulated dissolution profiles consisted of determining the geometric ratio of C_max_, AUC_0_^∞^, AUC_0_^Cmax,^ and T_max_ (kinetic parameters) of the generic/reference drugs, whereby generic drugs “C” and “D” presented the highest probability of similarity since their 90% confidence intervals were included, or they were very close to the acceptance interval (80.00–125.00%). These results were consistent with the *f*_2_, bootstrap *f*_2,_ and dissolution efficiency approaches (independent models). In conclusion, the proposed comparison method can be an important tool to establish similarity in dissolution profiles and to facilitate the development/selection of new formulations and positively ensure bioequivalence in clinical studies.

## 1. Introduction

Dissolution profiles are an essential tool to evaluate the in vitro release of one or more active pharmaceutical ingredients (APIs) from their solid dosage form [1]. This tool is widely used during the development of pharmaceutical products for the approval of generic drugs, and in addition, they play a fundamental role in the decision-making of regulatory entities since biowaivers can be obtained and demonstrate similarity despite post-approval changes to the drug [2]. However, to achieve the above, it is important to develop dissolution methods that are sufficiently discriminatory and robust so as not to overestimate results, and then, the selection of the dissolution apparatus, the agitation rate or work flow, the dissolution medium, and the analytical methodology are some of the critical factors to consider during their development [3].

In general, in the pharmaceutical industry, in vitro dissolution methods are performed in close-loop systems based on vessels, among which are the USP I (Baskets) and II (Paddles) apparatuses [4]. However, one of the disadvantages of the closed-loop systems of USP I and II apparatuses is that they can mask any slight differences in API release rates from the dosage forms, which can lead to inconsistencies between in vitro and in vivo tests [5]. In this sense, the flow-through cell system (USP IV apparatus) has currently gained more acceptance due to its versatility in dissolution tests, since it is a more discriminatory method due to its hydrodynamic conditions (laminar flow) [6]. The USP IV apparatus can be used in both open and closed-loop systems, according to the properties of the drug under study, but the open-loop system has the advantage of always keeping the dissolution medium fresh (sink conditions throughout the study), being able to obtain cumulative dissolution profiles from non-cumulative ones and providing an environment potentially closer to that of the gastrointestinal tract, making it useful for establishing in vitro and in vivo correlations (IVIVC) [7,8]. However, it is important to mention that one of its disadvantages is that for drugs with low solubility, it requires high volumes of dissolution medium [9].

Although the open-loop system of the USP IV apparatus has advantages over dissolution apparatuses that work in a closed-loop system, the dissolution profile comparison methods are designed for cumulative profiles. Then, the non-accumulated profiles obtained in the open-loop configuration of the USP IV apparatus must be transformed into accumulated profiles to be able to compare and establish similarity in test or generic drugs. These comparison methods consist of three groups: the methods based on analysis of variance (ANOVA), model-dependent methods, and model-independent methods, but the European Medicines Agency (EMA) and the U.S. Food and Drug Administration (FDA) guidelines on bioequivalence prefer the model-independent method based on the calculation of the similarity factor (*f*_2_) when the variability of the dissolution profiles complies with what is established in the guidelines [10]. However, when the variability in the dissolution profiles is high, there is no single comparison method, since the EMA prefers to use the bootstrap *f*_2_ approach and the FDA establishes the use of multivariate methods and dependent models, that is, still the comparison methods have not been homologated [1,11].

Therefore, the objective of the present study was to develop a discriminative dissolution method in the USP IV apparatus using the open-loop configuration, as well as to propose a method to compare non-cumulative dissolution profiles obtained in the open-loop configuration considering kinetic parameters and validate its predictive power through its comparison with independent and dependent immediate-release tablets of five commercial drugs (one reference drug and four generic drugs) of metoprolol tartrate as a model drug, which is a cardio selective beta-blocker drug that is used in the treatment of hypertension that belongs to group I of the biopharmaceutical classification system (solubility: >1000 mg/mL in water and LogP: 1.88) [12] and that according to the FDA requires bioequivalence studies to establish similarity, with doses of 20–100 mg daily being the most frequently used [13]. This API presents a pKa of 9.68, hydrogen bonding counts acceptor: 4, and donor: 2, and it is also chemically stable in acidic, neutral, and alkaline media [14].

## 2. Materials and Methods

### 2.1. Materials

The comparison of the dissolution profiles of metoprolol tartrate 100 mg immediate-release tablets was performed considering Lopresor^®^ 100 (“A”, Novartis Farmacéutica, Mexico City, Mexico, Lot N0059) as the reference drug and four generic drugs: Kenaprol^®^ (“B”, Laboratorios Kener, Mexico City, Mexico, lot M07410); Proken^®^ (“C”, Laboratorios Kendrick, Mexico City, Mexico, lot OJS957); Nipresol^®^ (“D”, Bruluart, Mexico City, México, lot 01052); and Metobest^®^ (“E”, Laboratorios Best, Mexico City, Mexico, lot 1009042). All medications were purchased at a local pharmacy. Metoprolol tartrate reference standard was purchased from the United States Pharmacopeia (99.7% on the as is basis, catalog no. 1441301, USP, St. Rockville, MD, USA). Hydrochloric acid (HCl) at 36.5–38.0% was purchased from JT Baker (JT Baker, by Fisher Scientific GmbH, Schwerte, Germany).

### 2.2. Dissolution Profile Studies 

#### 2.2.1. Test 1: Selection of Dissolution Media

The initial dissolution profiles were performed with six tablets of the reference drug “A” using the USP II apparatus (Vankel VK 7000, VanKel Industries, NJ, USA) at a stirring speed of 50 rpm and 900 mL of degassed simulated gastric fluid (without enzyme) or phosphate buffer pH = 6.8 as dissolution media at 37 °C ± 0.5 °C. Samples of 5 mL were withdrawn at 5, 10, 15, 20, 30, and 45 min without medium replacement, filtered through 0.45 μm Nylon Acrodiscs^®^ (Merck Millipore Ltd., Carrigtwohill, Ireland), and analyzed spectrophotometrically at 273 nm (UV-Vis Varian Cary 1E Spectrophotometer, Santa Clara, CA, USA) from a method previously validated by standard additions. The accumulated percentages of the dissolved drug were reported, considering the correction of the volume of the dissolution medium at each sampling time.

#### 2.2.2. Test 2: Dissolution Profiles in Apparatus II USP

Considering the results of test 1, test 2 was carried out with both the reference drug and the 4 generic drugs (n = 12). Dissolution profiles were performed using the USP apparatus II (Vankel VK 7000, VanKel Industries, Edison, NJ, USA) with 900 mL of degassed simulated gastric fluid (without enzyme) as the only dissolution media at 37 °C ± 0.5 °C and a stirring speed of 50 rpm. In each dissolution profile, samples of 5 mL were withdrawn at 2, 4, 6, 8, 10, 12, 14, 16, 18, 20, 25, 30, 40, 50, and 60 min without medium replacement, filtered through 0.45 μm Nylon Acrodiscs^®^ (Merck Millipore Ltd., Carrigtwohill, Ireland), and analyzed spectrophotometrically at 273 nm (Varian Cary 1E UV-Vis spectrophotometer, Santa Clara, CA, USA) in order to determine the amount of metoprolol tartrate dissolved at each sampling time. Corrections in the amount of dissolved metoprolol tartrate were made according to the volume setting at each sampling time.

#### 2.2.3. Test 3: Dissolution Profiles in Apparatus IV USP (Open-Loop Configuration)

Dissolution profiles of the drugs (reference and generics) were also obtained in a flow-through dissolution apparatus (Sotax CH-4008, Basel, Switzerland) equipped with 22.6 mm diameter cells. Briefly, a 5 mm diameter ruby bead was placed at the base of the 22.6 mm cell, followed by 3 gr of 3 mm diameter glass beads and a 2.7 μm Whatman^®^ glass microfiber filter (GF/D, Millipore-Sigma, St. Louis, MI, USA). The dissolution medium also comprised degassed simulated gastric fluid (without enzyme) at 37 °C, which was pumped at a flow rate of 8 mL/min. The dissolution apparatus was used in an open-loop configuration, considering 12 tablets for each product evaluated. Dissolution samples were collected manually every minute for 8 min, then every 2 min until reaching 20 min of accumulated dissolution, and subsequently, every 5 min until completing 40 min. Samples were filtered through 0.45 μm Nylon Acrodiscs^®^ (Merck Millipore Ltd., Carrigtwohill, Ireland) and analyzed spectrophotometrically at 273 nm (Varian Cary 1E UV-Vis spectrophotometer, Santa Clara, CA, USA), under the same conditions as in tests 1 and 2.

### 2.3. Similarity Evaluation

The dissolution profiles obtained in apparatus II (paddles) and IV (flow-through cell) USP were compared by independent and dependent approach models. Within the independent models, the calculation of the difference factor (*f*_1_), the similarity factor (*f*_2_), the derivation of the 95% confidence interval for *f*_2_ based on bootstrap (bootstrap *f*_2_), analysis of variance (ANOVA), and the dissolution efficiency (DE), as well as independent multivariate methods such as multivariate statistical distance (MSD) and time series approaches, were considered. In the case of the dependent models, the dissolution profiles were fitted to different mathematical models, but only the parameters of the best model were compared by MSD to determine similarity as recommended by the FDA guidelines [15]. Finally, the non-accumulated profiles obtained in the apparatus IV USP were the only ones that were subject to comparison from an independent model based on the calculation of kinetic parameters. All calculations were performed in both Microsoft™ Excel and Statgraphics Centurion XV (2007; Statistical Graphics Co., Rockville, MD, USA).

#### 2.3.1. *f*_1_, *f*_2,_ and Bootstrap *f*_2_ Approaches

The *f*_1_ and *f*_2_ were calculated considering the average values from the first sampling time to a maximum sampling time after one of the drugs (reference or generics) has reached 85% dissolved metoprolol tartrate. While a calculated value of *f*_1_ in the range of 0 to 15 suggests the similarity of the dissolution profiles between the drugs (Reference and generic), values of *f*_2_ greater than 50 (50–100) also suggest the similarity of the two profiles [1,16,17].

The following equations describe *f*_1_ and *f*_2_:$$f_{1}=\left[\overset{n}{\underset{t=1}{\sum }}\frac{\left|R_{t}-T_{t}\right|}{\sum_{t=1}^{n}R_{t}}\right]\times 100\;f_{2}=50\times \log_{10}\left[\frac{100}{\sqrt{1+\frac{\sum_{t=1}^{n}{\left(R_{t}-T_{t}\right)}^{2}}{n}}}\right]$$
where n is the number of sampling points, and R_t_ and T_t_ are the mean dissolution values of the reference drug and generic drug, respectively, at time t.

For the *f*_2_ bootstrap approach, the dissolution profile data sets were created by random sampling of individual dissolution rates at each time point in the original data, considering the criteria established above for *f*_2_. The number of randomly obtained dissolution profiles was set at 10,000, which was sufficient to stabilize the results. The similarity of the dissolution profiles was established considering the calculation of the 5th quartile of the *f*_2_ distribution, which must be greater than 50.

#### 2.3.2. ANOVA-Based Method

For this method, the percentages dissolved (dependent variable) at each sampling time (repeated factor) for the reference drug and the generic drugs were compared using a one-way ANOVA, followed by the construction of 95% confidence intervals for the difference of the drug dissolved means at each sampling time, according to the following expression:$$\left({\bar{y}}_{1}-{\bar{y}}_{2}\right)\pm t_{0.95,\mathrm{df}}\times \sqrt{\mathrm{MSE}\times \left(\frac{1}{n_{1}}+\frac{1}{n_{2}}\right)}$$
where y_1_ − y_2_ is the difference in the means of drug dissolved in the generic and reference drugs at each sampling time; df: degree of freedom; MSE: mean square of the error; and n_1_ and n_2_: number of samples of each drug.

In addition, drugs, time, and drug–time interaction (Drug × Time) were considered as the source of variation. As a criterion to establish similarity between the dissolution profiles, it was considered that the maximum difference of the acceptance limits in the percentages dissolved at each sampling time was ±10% according to the suggestions of the EMA guidelines for alternative methods of comparison [16].

#### 2.3.3. Dissolution Efficiency

The dissolution efficiency (DE) is defined as the area under the dissolution curve up to a certain time, t, which is calculated using the trapezoid method [18,19]. DE was calculated according to the following expression:$$\mathrm{DE}=\left\{\left[\int_{t_{1}}^{t_{2}}{\%D}_{t}\cdot \mathrm{dt}\right]/\left[{\%D}_{\max}\cdot (t_{2}-t_{1})\right]\right\}\cdot 100=\left[{\mathrm{AUC}}_{0-T}/{\%D}_{\max}\cdot T\right]\cdot 100$$
where %D_t_ is the percentage dissolved at time t, %D_max_ is the maximum percentage dissolved at the final time T, and AUC_0–T_ is the area under the curve from zero to T. The DE obtained from each of the individual tablets of both the reference drug and the generic drugs were compared by ANOVA and using 90% confidence intervals for the ratio of the means (log-transformed). The similarity criterion was based on the maximum limits of differences between the profiles (±10%), the same criterion as for the ANOVA-based method.

#### 2.3.4. Multivariate Statistical Distance (MSD)

The comparison of the dissolution profiles was also performed by the method of the maximum multivariate statistical distance (MSD) [20]. However, it is convenient for this type of method to investigate the symmetry of the variance–covariance matrix (homogeneity) so as not to overestimate the results from a 95% confidence chi-square test. Then, for variance–covariance matrices that were symmetric (homogeneous), the statistical methodology consisted of calculating the multivariate measure called the Mahalanobis distance (MD), which considered a vector of differences of the arithmetic average of the generic drug, with respect to the reference, at the different sampling times as well as the variance–covariance matrix of the dissolved percentages of both drugs at different times. Subsequently, the global similarity limits were established based on the average dissolved percentages of the reference drug with a tolerance of ±10% for MD (DM10%), followed by the similarity limits, which were calculated using the global similarity limits to establish the upper limit at a 90% confidence level for the true MD. The similarity in the dissolution profiles was considered when DM was less than the DM10% limit [11,21]. In contrast, when the variance–covariance matrix was not symmetric (heterogeneous), the method to evaluate the similarity of the dissolution profiles was through Hotelling’s T^2^ statistic (global similarity) and considering their 90% confidence intervals for each sampling time (local similarity) [22]. In this case, to declare the similarity between the dissolution profiles, the 90% confidence intervals of Hotelling’s T^2^ at each sampling time had to be included in the acceptance limit of ±10% [23]. The comparison of the dissolution profiles by this method was performed considering a point after one of the drugs (reference or generic) reached 85% dissolved, that is, the same criteria as for *f*_2_.

#### 2.3.5. Time Series Approach

This method considers the relative ratio between the dissolved percentage per tablet of the generic drug and the reference drug at each sampling time [24]. If this relative ratio approaches unity, then the dissolution profiles can be considered similar. However, it is important to determine if the relative ratios at the different sampling times of both drugs are homogeneous or heterogeneous to establish overall similarity, since the statistical analysis is different. Then, to determine the overall similarity between drugs, a 95% confidence interval was generated for the true value of the ratio of mean dissolution rates [lower limit (%LI) and upper limit (%UL)], which was compared with a lower (δ_L_) and upper (δ_U_) equivalence limit based on the Q value (% drug dissolved in a determined time “t”) applicable to the drug according to the monographs established in the pharmacopeias, in this case, Q = 75%. If the ratio of the mean dissolution rates was within the bioequivalence limit (δ_L_, δ_U_), then the dissolution profiles could be said to be similar. To evaluate the local similarity, the construction of confidence intervals for the relative dissolution ratios at each sampling moment was considered, and the same criteria were used to establish similarity between the profiles as for the global similarity.

#### 2.3.6. Dependent Model: Fit to Mathematical Models

The cumulative dissolution profiles of the reference and generic drugs obtained in both dissolution apparatus were adjusted to different mathematical models. Within the dependent models with a single parameter were found the first-order, second-order, Higuchi, Hixson–Crowell, and Korsmeyer–Peppas models (Table 1), which were used to determine dissolution rates and the release mechanism [25]. In the case of dependent models with two parameters, the Weibull, Gompertz, and logistic models were considered. In the case of the dependent models with two parameters, the Weibull, logistic, and Gompertz models (Table 1) were considered to establish the similarity of the dissolution profiles from MSD [19,26,27]. The selection of the best fit model was based on the highest coefficient of determination (r^2^) and the lowest Akaike information criterion (AIC) [28]. The comparison of the dissolution profiles obtained in both dissolution apparatus considering these methods consisted in the estimation of the 90% confidence region limits of their α (scale factor) and β (shape factor) parameters (log-transformed) and the region of similarity, which considered 2 standard deviations (2 STD) for the reference product, that is, approximately 95% confidence [15,19,27]. Dissolution profiles were considered similar if the 90% confidence region limits for each model parameter were included in the region of similarity [11].

#### 2.3.7. Kinetic Parameters: Non-Cumulative Dissolution Profiles

For this independent model, kinetic parameters such as maximum concentration (C_max_), time to reach maximum concentration (T_max_), area under the curve extrapolated from time zero to infinity (AUC_0_^∞^), and area under the curve from time zero to C_max_ (AUC_0_^Cmax^) were determined from non-cumulative dissolution profiles obtained in the USP IV dissolution apparatus (open-loop configuration). C_max_ and T_max_ were determined directly from the non- accumulated dissolution profiles. Meanwhile, AUC was calculated using the linear trapezoidal method (linear up and down) [29]. To determine the equivalence of the dissolution profiles of the reference drug with the generic drugs, 90% confidence intervals were constructed considering the % ratio of the geometric means (generic/reference) of the kinetic parameters (C_max_, AUC_0_^∞^, AUC_0_^Cmax,^ and T_max_) with the help of the Statgraphics XVI centurion software (Statpoint Technologies INC., Warrenton, VA, USA). If the confidence intervals in both drugs (reference and generic) were between 80.00% and 125.00%, the drugs were considered equivalent [16].

## 3. Results and Discussion

### 3.1. Selection of Dissolution Media (Test 1)

Dissolution profiles are an essential tool for evaluating the in vitro release of one or more active pharmaceutical ingredients (APIs) from their solid dosage form [1]. In the pharmaceutical industry, the comparison of dissolution profiles is used to evaluate the similarities in the formulations proposed during the development stages of a new generic drug in order to select the formulation with the greatest probability of success in a bioequivalence study [30], as well as to determine if the changes implemented at the formulation or process level, generally minor/moderate, affect the dissolution profile [31,32]. However, the development of a method for dissolution profiles that is sufficiently discriminative is not easy since the dissolution conditions must be appropriately selected, starting with the dissolution equipment, the agitation rate or flows, and the dissolution media (degassed), given that the dissolution methods described in the monographs of the United States Pharmacopeia (USP) or European Pharmacopeia (Ph. Eur.) are designed only to discriminate between variations of the critical quality attributes of the drugs [33].

Considering this background, the selection of the dissolution medium is one of the most important factors, since it has been described that the dissolution apparatus I (Baskets) and II (paddles) are the most used because they are simple, robust, well-standardized, and easy to operate [34]. In this sense, the selection of the medium during the development of a dissolution method should stop being arbitrary [35] and focus on the physicochemical properties of the API (solubility to maintain sink conditions) [36], as well as on the pH conditions prevailing in the gastrointestinal tract [37] and stability of the drug [3]. As can be seen in Figure 1, the dissolution behavior of the reference drug “A” in different media (degassed simulated gastric fluid (without enzyme) or phosphate buffer pH = 6.8) was very similar, so the criterion that defined the selection of degassed simulated gastric fluid (without enzyme) in the present study was based on the pharmacokinetic parameters reported after oral administration of 100 mg of metoprolol tartrate (Lopresor 100), focusing specifically on the time in which the maximum plasma concentration is reached (T_max_ = 1.63 ± 0.47 h) [38]. This T_max_ value indicates that the dissolution process is carried out rapidly in the stomach and immediately after gastric emptying the absorption process begins [39,40,41], that is, a dissolution medium of phosphate buffer pH = 6.8 was not representative.

### 3.2. Dissolution Profiles Obtained in the USP II Apparatus (Test 2)

Considering the results of test 1, test 2 was performed in the USP II apparatus using degassed simulated gastric fluid (without enzyme) as the only dissolution medium. Differences were observed in the dissolution behavior of the generic drugs compared to the reference drug (Figure 2). In fact, the generic drug “B” statistically presented the highest dissolution rate with respect to the reference drug “A” (*p* < 0.05). Meanwhile, generic drugs “D” and “E” statistically showed the slowest dissolution rate compared to the reference drug “A” (*p* < 0.05).

### 3.3. Dissolution Profiles Obtained in the USP IV Apparatus (Test 3)

Despite the marked differences in the dissolution profiles in test 2 (USP II apparatus), in test 3, the differences between the cumulative dissolution profiles of the different drugs obtained in the USP IV apparatus were more evident (Figure 3). However, the dissolution behavior was very similar to that observed in the dissolution profiles obtained in the USP II apparatus, since the generic drug “B” continued to be the one that statistically presented the highest dissolution rate compared to the reference drug “A” (*p* < 0.05). In contrast, generic drugs “C”, “D”, and “E” presented statistically lower dissolution rates compared to reference drug “A” (*p* < 0.05). These results can be supported by the fact that the USP IV apparatus is a more discriminating and versatile dissolution apparatus because its hydrodynamic is different compared to other dissolution apparatuses; that is, it is more efficient and reproducible since it avoids the formation of quiet zones due to the fact that the workflows, laminar or turbulent, are constant [6,42,43,44]. One of the great advantages of the USP IV apparatus is that from the open-loop configuration, not only non-cumulative dissolution profiles can be obtained, but these can be transformed into accumulated dissolution profiles, and finally be compared by dependents or independent methods to assess similarity [45].

### 3.4. Similarity Evaluation

The guidelines to establish similarity in dissolution profiles have not been homologated so far by the different most important regulatory entities in the world, so they are ambiguous and/or contradictory, especially in the context of highly variable dissolution profiles (>20% in the first time and >10% at subsequent sampling points) [32]. Next, different regulatory and non-regulatory approaches are presented to establish similarity in accumulated dissolution profiles obtained in the USP II and IV apparatuses, and in addition, the use of kinetic parameters is proposed to determine similarity in non-accumulated dissolution profiles obtained in the USP IV apparatus in an open-loop configuration.

#### 3.4.1. *f*_1_, *f*_2_, and Bootstrap *f*_2_ Approaches

Among the methods that can be used to compare dissolution profiles is the similarity factor (*f*_2_) approach, a relatively simple method that is vital for regulatory authorities as it requires very little statistical consideration in terms of dissolution data and calculations [17,46]. However, the rules and criteria associated with the application of this dissolution profile comparison method are not globally harmonized, but an *f*_2_ greater than 50 indicates that there is less than a 10% difference between the compared dissolution profiles. Another method of dissolution profile comparison established only by FDA guidelines is the difference factor (*f*_1_), which suggests a similarity between dissolution profiles when presenting values between 0 and 15 [17,47]. Both methods are used to compare dissolution profiles with low variability.

In the present study, the comparison of the dissolution profiles of generic drugs with respect to the reference drug using *f*_1_ and *f*_2_ is shown in Table 2. In general, the *f*_2_ values were lower in dissolution profiles obtained in the USP IV apparatus compared to those of the USP II apparatus, except for the generic drug “D” since the *f*_2_ value in the USP IV apparatus was slightly higher compared to that in the USP II apparatus. In this sense, while in the USP II apparatus the generic drug “B” was the only one that did not meet the criteria established for *f*_2_ (42.76), in the USP IV apparatus, generic drugs “B” and “E” presented *f*_2_ values of 36.58 and 46.46, respectively. In contrast, *f*_1_ values were higher in the dissolution profiles obtained in the USP IV apparatus, and even generic drugs B, C, and E did not meet the similarity criteria. In the case of the dissolution profiles obtained in the USP II apparatus, generic drug B was the only one that did not meet the similarity criteria (*f*_1_ = 30.65). This trend can again be supported since the USP IV apparatus presented a greater discriminative capacity to determine differences between dissolution profiles, which makes it an attractive dissolution apparatus for the development/selection of formulations and thus positively ensure clinical bioequivalence studies [7,48]. However, because the dissolution profiles of the reference drug “A” and the generic drugs “B” and “E” were of high variability, the comparison of profiles by the *f*_1_ and *f*_2_ approaches to establish similarity was not appropriate since their ability to identify real differences is limited [21].

In this sense, the EMA guidelines suggest that in order to compare highly variable dissolution profiles, the bootstrap methodology should be used to derive confidence intervals for *f*_2_ based on quantiles of resampling distributions, which contrasts with what is established in the FDA guidelines, since these suggest the use of multivariate methods, specifically the calculation of the multivariate statistical distance (MSD) [1]. In accordance with the above, it has been described that the *f*_2_ bootstrap method offers several advantages since its interpretation is the same as *f*_2_ and, moreover, it is more sensitive to detecting differences in dissolution profiles than multivariate methods based on the 90% confidence region of the Mahalanobis distance (MD) [49]. Table 2 shows that the bootstrap *f*_2_ values for the comparison of dissolution profiles were slightly lower than those of *f*_2_ in both dissolution apparatus, but the same trend was maintained in terms of similarity, that is, the dissolution profiles of generic drug B obtained in the USP II apparatus, and the dissolution profiles of generic drugs B and E obtained in the USP IV apparatus were not similar. 

#### 3.4.2. ANOVA-Based Method

ANOVA-based methods are not mentioned in any of the FDA or EMA guidelines. However, the EMA guidelines establish that when alternative comparison methods are used in dissolution profiles with high variability, the similarity acceptance limits should be predefined, justified, and not exceed a 10% difference. Likewise, for this specific comparison method, the variability in the dissolution profiles of the test and reference products should be similar, although less variability of the test product may also be acceptable [1,16]. Table 3 shows the results of the comparisons of the dissolution profiles between the reference drug and the generic drugs obtained in the USP II and IV apparatuses by the ANOVA-based method. The similarity between the dissolution profiles obtained in the USP II apparatus was established only between the reference drug “A” and the generic drug “C” since the 95% confidence intervals for the difference of the drug dissolved means at each sampling time were included in the acceptance limits of ±10%, regardless of whether the dissolution profiles obtained in the USP II apparatus complied with the variability established by the different regulatory entities. The other dissolution profiles obtained in the USP II and IV apparatuses were not similar, which contrasts with the results of *f*_1_, *f*_2,_ and bootstrap *f*_2_. These results were due to the fact that the ANOVA-based method is more sensitive than other comparison methods since it detects differences between the dissolution profiles in terms of level and shape [50,51,52]. Furthermore, it has been described that this dissolution profile comparison method violates the underlying assumption of independence because it does not consider the correlation between the dissolution data over time, and therefore it is not recommended [53]. However, its interest applies to immediate-release systems when it is required to compare a single point in dissolution to study its repeatability and reproducibility considering the different sources of variation that can affect the test [51,53].

#### 3.4.3. Dissolution Efficiency

Similar to the ANOVA-based method, dissolution efficiency (DE) is not considered a method of choice to establish similarity between dissolution profiles according to the FDA and EMA guidelines, but it can be used on highly variable dissolution profiles as long as it is statistically valid and satisfactorily justified [1]. Although the DE is mainly used to compare drug release [54], this method was proposed in 1975 to establish similarity mainly for immediate-release dosage forms [55]. Meanwhile, the median dissolution time (MDT) and median residence time (MRT) are more applicable to controlled release systems [56], and for this reason, they were not included in the analysis. Table 4 shows the results of the 90% confidence intervals for the DE ratio between generic drugs and the reference drug, as well as the decision that corresponds to the similarity between dissolution profiles considering 90.0–110.0% as acceptance limits, that is, a maximum difference of ±10%. In the case of the dissolution profiles obtained in the USP II apparatus (low variability), generic drug B was the only one considered not similar, which is consistent with the *f*_1_, *f*_2,_ and bootstrap *f*_2_ approaches. Instead, in the dissolution profiles obtained in the USP IV apparatus, which were characterized by high variability (reference drug “A” and generic drugs “B” and “E”), generic drug D was the only similar one, which is consistent with the *f*_1_ approach and to a certain extent with the *f*_2_ and bootstrap *f*_2_ approaches since generic drug “C” is at the limit of being able to be considered similar with the DE approach (88.93–91.47%), but it is also in the limit of being considered not similar by the *f*_2_ (53.78) and bootstrap *f*_2_ (50.81) approaches. In this sense, DE can be a good method for comparing dissolution profiles with both low and high variability.

#### 3.4.4. Multivariate Statistical Distance (MSD)

As previously mentioned, the FDA guidelines prefer the use of MSD methods considering data from dissolution profiles or on parameters of dependent models when the provisions of the different regulatory entities regarding the coefficient of variation at each sampling time are not met (>20% in the first time and >10% at subsequent sampling points), but the suggested recommendations are too general [1,15]. Thus, to carry out these types of comparison methods, the FDA guidelines established the calculation of the Mahalanobis distance (MD) as the method of choice, but this method assumes that the underlying variance–covariance matrix of the data from the dissolution profiles of the reference and generic drugs is symmetrical [53]. However, it has been described that when the variance–covariance matrix is not symmetric, Hotelling’s T^2^ test should be used [53]. Considering these assumptions, when comparing the dissolution profiles of the reference drug with the generic drugs obtained in both dissolution apparatus, Hotelling’s T^2^ test was used to establish similarity since the variance–covariance matrices were not symmetrical in any of the cases according to a chi-square test performed (*p* < 0.05). Table 5 shows the results of local similarity for the comparison of dissolution profiles obtained in the USP II apparatus (low variability). The only profile that met the similarity criteria was generic drug “C” since the 90% confidence intervals of Hotelling’s T^2^ at each sampling point were included in the acceptance limit of ±10%. Meanwhile, in the comparison of dissolution profiles obtained in the USP IV apparatus (high variability), none of the generic drugs presented similarity (Table 6) given that at least one of the 90% confidence intervals of Hotelling’s T^2^ were outside the acceptance limit of ±10%. These results are consistent with those of the ANOVA-based comparison method. Although it has been described that multivariate methods such as MSD are less discriminative and sensitive than the calculation of *f*_2_ [21] and that even a crossover in the dissolution profiles could have important implications in the similarity (non-discriminative, low specificity, and positive predictive value due to false positives) [10], these limitations can be solved to some extent if the comparison of dissolution profiles by this method is performed considering a point after reaching 85% dissolved, as described in the FDA and EMA guidelines for the calculation of *f*_2_ since the method becomes more sensitive to detect differences, and thus the dissolution profiles obtained in USP II and IV apparatuses were analyzed up to 25 min and 40 min of dissolution, respectively. However, the bootstrap *f*_2_ method is considered a better alternative to dissolution profiles with high variability [49,57].

#### 3.4.5. Time Series Approach

This method is not mentioned in any FDA or EMA guidance documents, but it was proposed as an alternative method to assess both global and local similarity in dissolution profiles with high variability through correlation between consecutive time points to describe the ratio of the percentages released of the reference and test drugs [53]; however, this method has been described as lacking in its scientific justification [58]. Table 7 shows the results of both global and local similarity of the comparison of dissolution profiles between the test drug and the generic drugs considering 95% confidence intervals and the homogeneity of the relative ratio per tablet at the different sampling times. In the case of the dissolution profiles obtained in the USP II apparatus, the relative ratio per tablet between the reference drug “A” and the generic drug “B” was the only homogeneous one. Regarding the dissolution profiles obtained in the USP IV apparatus, the relative ratio per tablet between the reference drug “A” and the generic drugs “B”, “C”, and “D” were homogeneous. According to the results (Table 7), none of the comparisons of the dissolution profiles of the generic drugs obtained in both dissolution apparatuses met the similarity criteria, since the global and local confidence intervals were not included in the similarity limit (87.5–114.29%), which is derived from the desired mean dissolution rate of a drug (Q) established in the drug monograph in the pharmacopeia. Specifically, for metoprolol tartrate immediate-release tablets Q = 75% in 30 min [59], but when the information of Q is not available, it is suggested to use the mean of the reference product [24]. These results obtained from the similarity in the dissolution profiles were mainly due to the fact that this approach has greater discriminative power over other comparison methods [60]. However, this higher discriminative power can lead to Type I errors, that is, concluding that the dissolution profiles are significantly different when in fact they are similar [53]. Therefore, this method of comparing dissolution profiles with high and low variability is not the most appropriate and requires modifications, besides which its interpretation is difficult when there is a discrepancy between the results of global and local similarity [60].

#### 3.4.6. Dependent Models: Fit to Mathematical Models

Dissolution profiles provide useful information on the release pattern of APIs as well as release mechanisms of the API from the pharmaceutical dosage form [61]. Therefore, the dissolution profiles of the reference drug and the generic drugs obtained in the USP II and IV apparatuses were adjusted to mathematical models such as zero-order kinetics, first-order kinetics, Higuchi, Hixson–Crowell, and Korsmeyer–Peppas, the latter being useful to interpret the release mechanism of the API. The fit of the cumulative dissolution profiles to the different mathematical models obtained in the USP II and IV apparatuses is shown in Table 8. The results showed that the dissolution profiles of the reference drug “A” and the generic drugs “D” and “E” obtained in both dissolution apparatuses were adjusted to zero-order kinetic models since they presented the highest coefficients of determination (r^2^) and the lowest values in the Akaike information criterion (AIC); that is, the release of the API is a function of time and the process takes place at a constant rate independent of API concentration [62]. In addition, their release mechanisms were adjusted to super case-II transports since their diffusion exponents “n” were greater than 0.89 in the Korsmeyer–Peppas model, so the API diffusion process in the non-swellable matrix was very fast [63,64]. On the contrary, the dissolution profiles of the generic drugs “B” and “C” obtained in both dissolution apparatuses were adjusted to the Hixson–Crowell model (Table 8), and therefore the drug release was limited by the dissolution velocity and not by diffusion. However, as the diffusion coefficient “n” was less than 0.89 in the case of the dissolution profiles obtained in the USP II apparatus, the drug release mechanism through generic drugs “B” and “C” is governed by diffusion and swelling [63].

Although it has been reported that model-dependent approaches with a single parameter, such as those used above, can be subject to MSD analysis to establish similarities in dissolution profiles with high variability [11,15], the drawback was that the dissolution profiles obtained in both dissolution apparatuses of generic drugs “B” and “C” could not be compared since they were adjusted to different models compared to the reference drug “A”.

However, to solve this limitation, dependent models of two parameters were used as a method of comparing dissolution profiles to establish similarity, according to the recommendations of the FDA and some authors [15,65]. In this sense, the Weibull, logistic, and Gompertz models were used to fit the dissolution profiles of the reference and generic drugs, but the Weibull model was the only one that was used to establish the similarity of the dissolution profiles through the estimation of the 90% confidence region limits of its α (scale factor) and β (shape factor) parameters and the region of similarity considering two standard deviations (2 STD) for the reference drug since it was the model that presented the highest r^2^ and lowest AIC. The fit results were logical as the Weibull model has been reported to be the most flexible model for describing a wide variety of shapes compared to the other dependent models with two parameters [66]. Table 9 and Table 10 show the similarity results in the dissolution profiles obtained in USP II and IV apparatuses, respectively, through the comparison of the α and β parameters of the Weibull model. However, none of the dissolution profiles were similar, regardless of the dissolution apparatus used since in all cases the 90% confidence region limits of model parameters were outside the region of similarity. These results can be attributed to the multivariate method used, since as previously mentioned, these types of methods are less discriminative and more sensitive, although a potential danger has also been described for the model parameters since they can be biased or not be estimable if the sampling points are not chosen properly [66]. However, to minimize this last point, the comparison of the dissolution profiles obtained in the USP II and IV apparatuses were compared up to 25 min and 40 min, respectively. In short, these dependent models can be used more in the comparison of intra-batch dissolution profiles when there are minimal changes at the formulation level [27].

#### 3.4.7. Independent Model Comparison for Non-Accumulated Data (USP IV Apparatus)

Although some advantages of the USP IV apparatus have been previously described, its use specifically in the open-loop configuration has shown great potential for in vitro–in vivo correlations since information is lost when the closed-loop configuration is used, making the dissolution profile less discriminative. [5]. Likewise, it has been reported that the open-loop configuration allows for facilitating the reproduction of dissolution profiles when they are developed in independent laboratories [8]. Figure 4 shows the non-accumulated dissolution profiles of the drugs obtained in the USP IV apparatus using the open-loop configuration. The dissolution behavior of the generic drug “B” turned out to be very different compared to that of the reference drug “A” since its dissolution process was faster, and it reached higher concentrations of metoprolol tartrate. By contrast, generic drugs “C”, “D”, and “E” presented slower dissolution processes and therefore reached lower concentrations of metoprolol tartrate compared to the reference drug “A”. Even though the comparison methods for dissolution profiles with low and high variability are designed for cumulative profiles according to the different regulatory entities, they can underestimate or overestimate the similarity between them since they are not homologated at the regulatory level or because they are too sensitive, causing a decrease in discriminative power. Therefore, the preference for the dissolution profile comparison method with high variability will depend on the regulatory entity. However, it is important to consider the fields of application of the dependent and independent models suggested above, as well as their degree of complexity, which could be a limitation for their use.

In this sense, in the present study, the comparison of kinetic parameters (C_max_, AUC_0_^∞^, AUC_0_^Cmax^ and T_max_) obtained from non-accumulated dissolution profiles was proposed to establish similarities, as is undertaken for bioequivalence studies, following the FDA and EMA guidelines [16,67]. The results in Table 11 show that the generic drugs “B” and “E” presented the highest and lowest C_max_, respectively, compared to the reference drug “A”. Regarding the T_max_ in which C_max_ was reached, the generic drug “C” presented the smaller value (6.00 min), followed by the generic drug “B” (6.42 min) when compared to the reference drug “A” (7.25 min). In contrast, the generic drugs “D” and “E” showed the highest T_max_, 8.00 min and 10.33 min, respectively. Furthermore, although the AUC_0_^∞^ did not show differences between the drugs, the AUC_0_^Cmax^ of the generic drugs “B”, “D”, and “E” was higher compared to that of the reference drug “A”. Instead, the generic drug “C” had a lower AUC_0_^Cmax^ in comparison with the reference drug “A”.

The comparison of the kinetic parameters showed that there were significant differences between the dissolution profiles. However, according to bioequivalence guidelines [16,67], to determine if a product is bioequivalent, in this case “similarity”, 90% confidence intervals were calculated considering the % ratio of the geometric means (generic/reference) of C_max_, AUC_0_^∞^, AUC_0_^Cmax,^ and T_max_, but only AUC_0_^Cmax^ and C_max_ (in this order) were considered to establish similarity between the dissolution profiles, as long as their confidence intervals were within the acceptance interval of 80.00 to 125.00% [4,68], in the case of immediate-release drugs. Table 11 also shows the % ratio of the geometric means of the kinetic parameters (C_max_, AUC_0_^∞^, AUC_0_^Cmax^, and T_max_) with their respective 90% confidence intervals. The % ratio of the geometric means of AUC_0_^Cmax^ and C_max_ for the generic drug “C” (C/A, AUC_0_^Cmax^: 85.04–93.95% and C_max_: 75.25–81.97%) and “D” (D/A, AUC_0_^Cmax^: 125.12–136.84% and C_max_: 79.71–87.05%) were the ones that came closest to the acceptance interval (80.00 to 125.00%), so the probability that they are similar with the reference drug “A” is high and is also supported by the values of *f*_2_ and bootstrap *f*_2_ since in both drugs it was greater than 50 and, in addition, it is related to the method of DE since generic drug “D” was considered similar and generic drug C was very close to being considered similar in the dissolution profiles obtained in the USP IV apparatus. In the case of the generic drug “E”, while the interval of the % ratio of the geometric means of AUC_0_^Cmax^ (105.88–131.03) was slightly above those of acceptance, the interval of the % ratio of the geometric means of C_max_ (64.41–72.59%) was the lowest. Finally, the intervals of the % ratio of the geometric means of AUC_0_^Cmax^ and C_max_ for the generic drug “B” (B/A, AUC_0_^Cmax^: 148.03–192.01% and C_max_: 143.01–166.46%) were very high. Therefore, the probability of similarity of generic drugs “B” and “E” with the reference drug “A” is supported by the fact that in both cases the values of *f*_2_ and bootstrap *f*_2_ were less than 50 and the DE method considered them not similar. In this sense, the analysis of kinetic parameters from non-cumulative dissolution profiles can be an excellent option to determine if the compared drugs are similar since the dissolution profiles obtained in the USP IV apparatus using the open-loop configuration are more uniform and discriminative compared to those of the USP I and II apparatuses [4,69]. Therefore, the USP IV apparatus in the open-loop configuration can be an excellent tool to ensure similarity in dissolution profiles during the drug development stage, as well as to detect variations at the formulation level when there are minor changes and evaluate the variability per dosage unit (process control).

## 4. Conclusions

The method to compare dissolution profiles of metoprolol tartrate immediate-release tablets in the USP apparatus IV was found to be more discriminative compared to that in the USP apparatus II. However, the open-loop configuration of the USP IV apparatus has more advantages compared to the closed-loop configuration since non-cumulative profiles can be converted to cumulative profiles to assess similarity across both independent and dependent models. However, in the present study, an alternative method for comparison of non-cumulative dissolution profiles using generic/reference kinetic parameter ratios (C_max_, AUC_0_^∞^, AUC_0_^Cmax^ and T_max_), a method that resembles the comparison of pharmacokinetic parameters obtained in bioequivalence studies, was proposed and was found to be consistent with the *f*_2_, bootstrap *f*_2,_ and DE approaches, since from these, similarity was established in the generic drugs “C” and “D” and not similarity in the generic drugs “B” and “E”. Therefore, this type of comparison method can be an important tool to facilitate the development/selection of new formulations and positively ensure clinical bioequivalence studies. Therefore, the proposed comparison method can be an important tool to establish similarity in dissolution profiles and to facilitate the development/selection of new formulations and positively ensure bioequivalence in clinical studies. Furthermore, it is a method that does not require complex calculations such as those required by multivariate methods and special conditions of analysis to avoid overestimating the results.

## Figures and Tables

**Figure 1 pharmaceutics-15-02191-f001:**
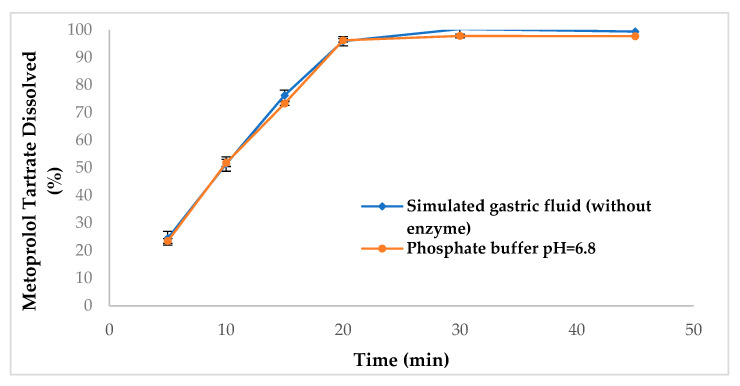
Dissolution profiles of metoprolol tartrate 100 mg immediate-release tablets in simulated gastric fluid (without enzymes) and phosphate buffer pH 6.8 using the USP dissolution apparatus II (Paddle) at 50 rpm and 37 °C. The results are expressed as mean ± SD (n = 6).

**Figure 2 pharmaceutics-15-02191-f002:**
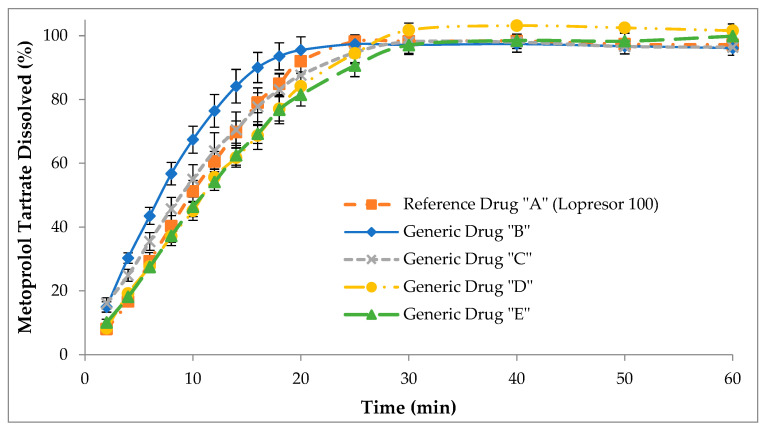
Cumulative dissolution profiles of reference drug “A” (Lopresor 100) and generic drugs “B”, “C”, “D”, and “E” obtained using the USP II apparatus. The results are expressed as mean ± SD (n = 12).

**Figure 3 pharmaceutics-15-02191-f003:**
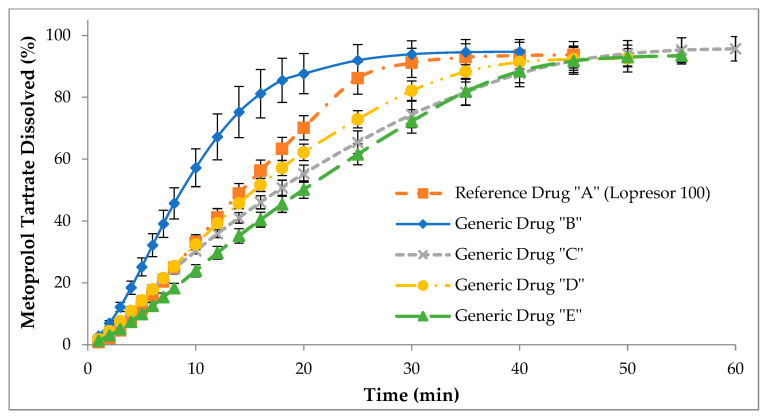
Cumulative dissolution profiles of reference drug “A” (Lopresor 100) and generic drugs “B”, “C”, “D”, and “E” obtained using the USP IV apparatus. The results are expressed as mean ± SD (n = 12).

**Figure 4 pharmaceutics-15-02191-f004:**
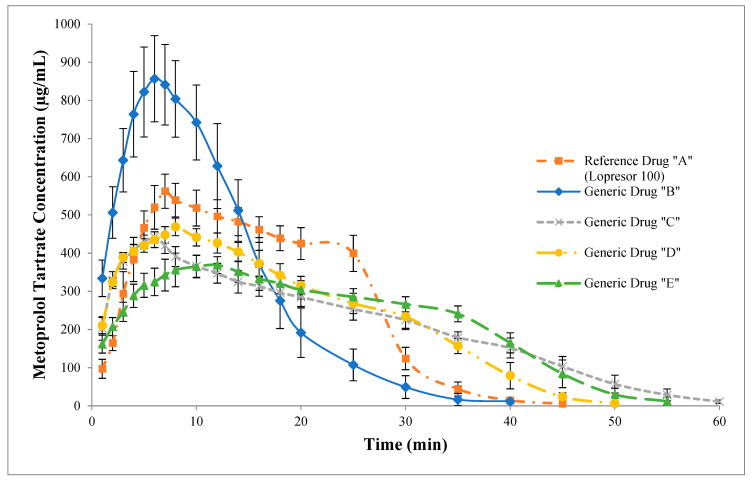
Non-cumulative dissolution profiles of reference drug “A” (Lopresor 100) and generic drugs “B”, “C”, “D”, and “E” obtained using the USP IV apparatus. The results are expressed as mean ± SD (n = 12).

**Table 1 pharmaceutics-15-02191-t001:** Mathematical models for adjustment in dissolution profiles.

Model	Equation
Zero-order	$Q_{t}=Q_{0}+K_{0}t$
First-order	${\mathrm{In}Q}_{t}={\mathrm{In}Q}_{0}-K_{1}t$
Higuchi	$Q_{t}=K_{H}\cdot \sqrt{t}$
Hixon–Crowell	$Q_{0}^{1/3}-Q_{t}^{1/3}=K_{s}\cdot t$
Korsmeyer–Peppas	$Q_{t}/Q_{\infty }=K_{k}\cdot t^{n}$
Weibull	$Q_{t}=Q_{\max}\cdot \left\{1-\exp\left[-a\cdot {\left(t\right)}^{b}\right]\right\}$
Gompertz	$Q_{t}=Q_{\max}\cdot \exp\left\{-a\cdot \exp\left[-b\cdot \log\left(t\right)\right]\right\}$
Logistic	$Q_{t}=Q_{\max}\cdot \left\{\frac{\exp\left[a+b\cdot \log\left(t\right)\right]}{1+\exp\left[a+b\cdot \log\left(t\right)\right]}\right\}$

Q_t_: amount of drug release in time t; Q_0_: initial amount of drug in the tablet; Q_t_/Q_∞_: fraction of drug release at time t; Q_max_: maximum dissolved at the final time; k_0_, k_1_, k_H_, k_k_, k_s_: release rate constants; n: release exponent; and a and b: parameters of the models.

**Table 2 pharmaceutics-15-02191-t002:** Comparison of the difference factors (*f*_1_), similarity factors (*f*_2_), and bootstrap *f*_2_ of the dissolution profiles obtained in apparatus II and IV USP of the generic drugs with the reference drug “A” (Lopresor 100).

Drug	Difference Factor (*f*_1_)		Similarity Factor (*f*_2_)		Bootstrap *f*_2_	
	USP Dissolution Apparatus					
	II	IV	II	IV	II	IV
Generic Drug “B”	**30.65 ***	**58.78**	**42.76**	**36.58**	**39.84**	**34.60**
Generic Drug “C”	8.13	**18.32**	64.60	53.78	59.96	50.81
Generic Drug “D”	9.89	10.99	60.39	65.30	55.25	61.05
Generic Drug “E”	10.40	**26.99**	59.47	**46.46**	54.76	**44.09**

* Values marked in bold indicate that the criteria for establishing similarity are not met.

**Table 3 pharmaceutics-15-02191-t003:** Comparisons of dissolution profiles of generic drugs with reference drug “A” (Lopresor 100) obtained in the USP II and IV apparatus using ANOVA-based statistical analysis ^1^.

Drug	USP II Apparatus			USP IV Apparatus		
	Two-Factor *p* Value	95% CI	Decision	Two-Factor *p* Value	95% CI	Decision
Generic Drug “B”	<0.01	Significant differences at 4–16 min	Non-similar	<0.01	Significant differences at 4–20 min	Non-similar
Generic Drug “C”	0.10	No significant differences	Similar	<0.01	Significant differences at 16–35 min	Non-similar
Generic Drug “D”	<0.01	Significant differences at 16 min	Non-similar	<0.01	Significant differences at 35 and 30 min	Non-similar
Generic Drug “E”	<0.01	Significant differences at 16 and 20 min	Non-similar	<0.01	Significant differences at 10 and 35 min	Non-similar

^1^ *p* value represents the interaction between drug and time (D × T).

**Table 4 pharmaceutics-15-02191-t004:** Comparison of dissolution profiles obtained in the USP II and IV apparatus by 90% confidence intervals for the mean ratio of dissolution efficiency (DE %) between the generic drugs with respect to the reference drug “A” (Lopresor 100) (G/R).

Drug	USP II Apparatus		USP IV Apparatus	
	DE % 90% CI for Mean Ratio G/R	Decision ^1^	DE % 90% CI for Mean Ratio G/R	Decision
Generic Drug “B”	117.91 114.68–121.24	Non-similar	117.71 115.39–120.07	Non-similar
Generic Drug “C”	106.58 103.57–109.67	Similar	90.19 88.93–91.47	Non-similar
Generic Drug “D”	95.72 93.02–98.51	Similar	94.54 92.92–96.20	Similar
Generic Drug “E”	98.56 95.88–101.31	Similar	83.40 82.07–84.76	Non-similar

^1^ Similarity decision between the profiles (generic/reference) was made considering a maximum difference of ±10%, that is, considering a similarity interval between 90.0% and 110.0%.

**Table 5 pharmaceutics-15-02191-t005:** Multivariate analysis based on 90% confidence intervals for each sampling time of the reference drug “A” (Lopresor 100)/generic drug profiles obtained in the USP II apparatus using Hotelling’s T^2^ statistic (local similarity).

Sampling Time (min)	90% Confidence Intervals (90% CI)			
	USP II Apparatus			
	Generic Drug B	Generic Drug C	Generic Drug D	Generic Drug E
2	[5.65, 8.23]	[7.13, 9.03]	[−0.73, 1.23]	[1.02, 3.16]
4	[12.44, 14.81]	[7.16, 9.24]	[1.41, 3.64]	[−0.07, 2.96]
6	[12.74, 15.80]	[4.60, 7.91]	[−3.42, 0.20]	[−3.83, 0.34]
8	[14.42, 18.35]	[3.46, 7.26]	[−5.18, −1.29]	[−5.67, −0.58]
10	[13.94, 18.52]	[1.21, 6.42]	[−8.09, −4.11]	[−7.82, −1.82]
12	[13.65, 18.20]	[−0.03, 6.82]	[−6.70, −3.12]	[−9.15, −3,62]
14	[11.82, 16.93]	[−2.65, 4.11]	[−10.29, −5.73]	[−10.37, −4.16]
16	[8.83, 13.19]	[−4.01, 1.52]	[−12.91, −7.70]	[−12.55, −7.03]
18	[6.80, 10.45]	[−4.21, 0.57]	[−10.54, −5.12]	[−11.77, −4.42]
20	[1.99, 4.99]	[−6.50, −2.19]	[−10.56, −5.11]	[−13.81, −7.25]
25	[−2.56, 0.95]	[−5.50, −1.41]	[−5.68, −1.79]	[−9.55, −5.53]
Conclusion	Non-similar	Similar	Non-similar	Non-similar

**Table 6 pharmaceutics-15-02191-t006:** Multivariate analysis based on 90% confidence intervals for each sampling time of the reference drug “A” (Lopresor 100)/generic drug profiles obtained in the USP IV apparatus using Hotelling’s T^2^ statistic (local similarity).

Sampling Time (min)	90% Confidence Intervals (90% CI)			
	USP IV Apparatus			
	Generic Drug B	Generic Drug C	Generic Drug D	Generic Drug E
1	[1.66, 2.24]	[0.67, 1.05]	[0.75, 1.09]	[0.33, 068]
2	[4.15, 5.35]	[1.90, 2.49]	[1.95, 2.49]	[0.51,1.16]
3	[6.61, 8.62]	[2.35, 3.39]	[2.54, 3.45]	[−0.17, 1.01]
4	[9.38, 12.08]	[2.28, 3.78]	[2.53, 3.77]	[−1.19, 0.43]
5	[12.01, 15.24]	[1.96, 3.47]	[2.11, 3.42]	[−2.52, −0.75]
6	[14.38, 18.38]	[1.25, 2.86]	[1.33, 2.78]	[−4.27, −2.25]
7	[16.27, 21.05]	[−0.01, 1.76]	0.28, 1.92]	[−6.28, −3.90]
8	[18.08, 23.58]	[−1.33, 0.63]	[−0.43, 1.45]	[−8.02, −5.21]
10	[20.70, 27.41]	[−4.03, −1.68]	[−1.92, 0.44]	[−11.00, −7.51]
12	[22.05, 30.09]	[−6.65, −3.87]	[−3.29, −046]	[−13.41, −9.41]
14	[21.88, 30.81]	[−9.50, −6.17]	[−4.76, −057]	[−15.97, −11.39]
16	[20.66, 29.09]	[−12.11, −8.34]	[−6.28, −2.93]	[−18.28, −13.44]
18	[18.27, 26.07]	[−14.63, −10.50]	[−7.99, −4.37]	[−20.47, −15.37]
20	[13.77, 21.26]	[−17.07, −12.57]	[−9.88, −6.02]	[−22.68, −17.27]
25	[2.06, 9.49]	[−23.87, −17.56]	[−15.86, −10.73]	[−27.94, −21.25]
30	[−0.72, 6.25]	[−19.95, −13.46]	[−11.69, −6.39]	[−22.10, −15.81]
35	[−1.73, 5.04]	[−14.47, −8.15]	[−7.06, −2.26]	[−14.20, −8.02]
40	[−2.14, 4.67]	[−9.10, −2.56]	[−4.70, 0.43]	[−7.97, −2.26]
Decision	Non-similar	No-similar	Non-similar	Non-similar

**Table 7 pharmaceutics-15-02191-t007:** Comparison of dissolution profiles obtained in the USP II and IV apparatus by a time series approach considering 95% confidence intervals for global and local similarities.

Drug	USP II Apparatus			USP IV Apparatus		
	95% CI (Global Similarity)	95% CI (Local Similarity)	Decision	95% CI (Global Similarity)	95% CI (Local Similarity)	Decision
Generic Drug “B”	123.01–144.84	Significant differences before 18 min	Non-similar	157.67–215.20	Significant differences at all times of the profile	Non-similar
Generic Drug “C”	108.50–126.72	Significant differences before 12 min	Non-similar	97.10–129.22		Non-similar
Generic Drug “D”	82.11–107.90	Significant differences at 4 min and between 8–20 min	Non-similar	103.95–133.50		Non-similar
Generic Drug “E”	73.50–119.10	Significant differences at all times, except at 6 min	Non-similar	63.57–116.63		Non-similar
**Limit of similarity: 87.50–114.29% ^1^**						

^1^ Limit of similarity calculated considering that the metoprolol tartrate monograph at USP establishes a Q = 75%.

**Table 8 pharmaceutics-15-02191-t008:** Parameters of the mathematical models and descriptive statistics of regression for the dissolution data.

Model	Statistics ^1^	USP II Apparatus					USP IV Apparatus				
		Reference “A”	Generic “B”	Generic “C”	Generic “D”	Generic “E”	Reference “A”	Generic “B”	Generic “C”	Generic “D”	Generic “E”
Zero-order	r^2^	0.9922	0.9166	0.9502	0.9898	0.9929	0.9749	0.9657	0.9919	0.9953	0.9883
	k_0_ (%Dis ∗ min^−1^)	4.9024	5.9298	5.0479	4.4087	4.4161	3.3844	4.9917	2.8718	3.1668	2.4538
	MSE	5.8329	63.9021	28.3103	5.4318	3.7852	14.3348	32.9669	1.8453	1.9696	3.1387
	AIC	36.2512	57.9025	50.2955	33.8004	31.2852	74.3746	85.3320	48.9331	46.5559	53.3278
First-order	r^2^	0.9286	0.9612	0.9777	0.9602	96.3200	0.9144	0.9403	0.9842	0.9654	0.9583
	k_1_ (min^−1^)	0.0770	0.1136	0.0844	0.0660	0.0663	0.0447	0.0827	0.0370	0.0417	0.0300
	MSE	53.1229	31.1351	13.0971	22.6939	20.2464	48.8620	58.3842	5.1922	14.4517	11.2917
	AIC	56.4263	50.4021	42.4009	47.7845	47.2864	91.9112	92.9871	58.6913	74.7632	71.5421
Higuchi	r^2^	0.8133	0.9128	0.9075	0.8409	0.8451	0.7250	0.8315	0.8205	0.7920	0.7554
	k_H_ (%Dis ∗ min^−1/2^)	17.1046	21.0966	17.9065	15.4558	15.4901	11.5370	0.1756	10.0318	10.9832	8.4341
	MSE	138.6870	67.5427	51.8659	88.2738	83.5642	155.4990	156.9909	56.9645	85.8074	65.9340
	AIC	65.0834	58.3930	56.1195	60.8167	60.4724	108.4083	108.3192	94.2654	100.1338	96.4672
Hixson–Crowell	r^2^	0.9621	0.9908	0.9942	0.9814	0.9848	0.9377	0.9688	0.9940	0.9800	0.9704
	k_s_ (%Dis ∗ min^−1/3^)	0.0223	0.0313	0.0240	0.0194	0.0194	0.0137	0.0239	0.0114	0.0127	0.0094
	MSE	28.1260	7.5387	3.4102	10.7124	8.4516	35.5652	30.6325	2.6745	8.3986	7.9981
	AIC	50.6311	36.0721	30.1136	40.2526	38.9975	87.3931	83.4847	45.8812	67.0090	66.6547
Korsmeyer–Peppas	r^2^	0.9933	0.9847	0.9961	0.9945	0.9974	0.9936	0.9746	0.9954	0.9963	0.9975
	k_k_ (%Dis ∗ min^−n^)	5.0637	11.8948	9.1967	5.3606	5.4351	1.8844	6.6770	3.1345	2.8404	1.6715
	n	0.9920	0.7322	0.7703	0.9264	0.9217	1.2229	0.8935	0.9679	1.0423	1.1469
	MSE	5.6997	13.8821	2.6908	3.5050	1.6278	3.9871	26.6685	1.5489	1.6252	0.7375
	AIC	36.9205	43.8078	27.3260	28.0736	24.7320	56.6951	83.1922	44.7129	44.7520	30.4641

^1^ Coefficient of determination (r^2^); constants of the dependent models (k_0_, k_1_, k_H_, k_s_, k_k_); diffusion exponent (n); mean square error (MSE) and Akaike information criterion (AIC).

**Table 9 pharmaceutics-15-02191-t009:** Comparison of the fit parameters of the Weibull model for the dissolution profiles between the reference drug “A” (Lopresor 100) and the generic drugs obtained in the USP II apparatus through multivariate confidence regions.

Weibull Parameters	Ln Differences	USP II Apparatus			
		Generic Drug “B”	Generic Drug “C”	Generic Drug “D”	Generic Drug “E”
α ^1^	90% CI	−0.125 to −0.090	0.403 to 0.454	0.078 to 0.184	0.157 to 0.248
	2 STD Similarity region ^3^	−0.045–0.045			
β ^2^	90% CI	−0.189 to −0.072	−0.122 to −0.107	−0.071 to −0.037	−0.077 to −0.055
	2 STD Similarity region	−0.008–0.008			
Decision		Non-similar	Non-similar	Non-similar	Non-similar

^1^ *α*: scale factor, ^2^ *β*: shape factor and ^3^ 2 STD is approximately 95% confidence.

**Table 10 pharmaceutics-15-02191-t010:** Comparison of the fit parameters of the Weibull model for the dissolution profiles between the reference drug “A” (Lopresor 100) and the generic drugs obtained in the USP IV apparatus through multivariate confidence regions.

Weibull Parameters	Ln Differences	USP IV Apparatus			
		Generic Drug “B”	Generic Drug “C”	Generic Drug “D”	Generic Drug “E”
α ^1^	90% CI	0.484 to −0.654	0.374 to 0.521	0.333 to 0.461	0.164 to 0.319
	2 STD Similarity region ^3^	−0.093–0.0933			
β ^2^	90% CI	−0.084 to −0.048	−0.147 to −0.114	−0.118 to −0.092	−0.117 to −0.085
	2 STD Similarity region	−0.019–0.019			
Decision		Non-similar	Non-similar	Non-similar	Non-similar

^1^ *α*: scale factor, ^2^ *β*: shape factor and ^3^ 2 STD is approximately 95% confidence.

**Table 11 pharmaceutics-15-02191-t011:** Kinetic parameters of dissolution profiles obtained in the USP IV apparatus and their 90% confidence intervals (CI) of the ratio of geometric means (generic/reference).

Parameter	Geometric Mean ± SE					Geometric Point Estimate Ratio			
	(90% CI)					(90% CI)			
	Reference (A)	Generic (B)	Generic (C)	Generic (D)	Generic (E)	B/A	C/A	D/A	E/A
C_max_ (µg/mL)	565.21 ± 13.39	869.69 ± 31.73	441.77 ± 4.16	468.73 ± 6.65	384.54 ± 7.95	153.29	78.36	83.10	68.09
	(541.17–589.26)	(812.72–926.67)	(434.30–449.25)	(456.78–480.68)	(370.27–398.81)	(142.55–164.84)	(74.91–81.98)	(79.23–87.15)	(64.48–71.89)
AUC_0_^∞^ (µg·min/mL)	12,507.10 ± 189.35	12,580.20 ± 135.68	12,507.20 ± 151.82	12,234.30 ± 157.75	12,352.10 ± 115.31	101.57	99.19	97.86	98.84
	(12,167.10–12,847.10)	(12,336.50–12,823.90)	(12,234.50–12,779.80)	(11,951.00–12,517.60)	(12,145.00–12,559.20)	(98.61–104.63)	(96.12–102.36)	(97.61–98.11)	(95.21–102.62)
AUC_0_^Cmax^ (µg·min/mL)	2203.29 ± 57.32	3857.76 ± 185.18	1966.84 ± 20.74	2864.59 ± 22.51	2896.96 ± 188.61	173.57	89.52	131.09	128.99
	(2099.41–2307.18)	(3525.20–4190.33)	(1929.58–2004.09)	(2824.16–2905.02)	(2558.25–3235.68)	(157.88–190.82)	(85.11–94.17)	(124.82–137.68)	(114.23–145.67)
T_max_ (min)	7.25 ± 0.25	6.42 ± 0.19	6.00 ± 0.00	8.00 ± 0.00	10.33 ± 0.54	88.57	83.20	110.94	141.17
	(6.80–7.70)	(6.07–6.76)	(6.00–6.00)	(8.00–8.00)	(9.36–11.31)	(82.53–95.06)	(78.88–87.77)	(105.17–117.02)	(127.35–156.49)

C_max_: maximum concentration, T_max_: time to reach maximum concentration, AUC_0_^∞^: area under the curve extrapolated from time zero to infinity, and AUC_0_^Cmax^: area under the curve from time zero to C_max_.

## Data Availability

The data that support the findings of this study are available within this article, or from the corresponding author upon reasonable request.

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
