# Peer review of "Discriminative Dissolution Method Using the Open-Loop Configuration of the USP IV Apparatus to Compare Dissolution Profiles of Metoprolol Tartrate Immediate-Release Tablets: Use of Kinetic Parameters"

_pharmaceutics, 2023, doi:10.3390/pharmaceutics15092191_

Round 1

Reviewer 1 Report

This article presents a discriminative method for dissolution profiles to predict the pharmacokinetic behavior of commercial formulations of metoprolol as a model drug.

The introduction of the article reflects the considerations and problems that give rise to the development of the project, it is in accordance with the methodology and the results presented. Each of the methodologies used meets international requirements. They are all widely recognized, and in the same way, the statistical approaches that are evaluated here show an adequate application and directed specifically to the work developed.

The novelty of the work is wide, since the prediction of the bioavailability of drugs by means of in vitro tools is still a frontier issue, which is in full development and the use of the USP IV Dissolution Apparatus is one of the few ways of predicting bioavailability in vitro.

Author Response

The authors are very grateful for the comments and suggestions made to improve the quality of this manuscript before its possible publication.

Reviewer 2 Report

Comment

Date: 05-08-2023

Manuscript ID: pharmaceutics-2562994

Solis-Cruzet al. addressed very interesting findings in their research article entitle as “Discriminative dissolution method using the open-loop 2 configuration of the USP IV apparatus to compare dissolution 3 profiles of metoprolol tartrate immediate-release tablets: use of kinetic parameters”. The work is interesting, well designed, and informative to reader working in the domains. However, I recommend few suggestions before publication.

Comment 1: In the sentence “However, one of the disadvantages of the closed-loop systems of USP â…  and â…¡ apparatuses is that they can mask small differences in API release rates, which can lead to inconsistencies between in vitro and in vivo tests” the authors are recommended to explain the “small differences” generally encountered in the proposed study.  Moreover, the authors reported in the sentence as  “In this sense, currently, the flow-through cell system (USP IV apparatus) has currently gained more acceptance due to its versatility in dissolution tests, since it is a more discriminatory method due to its hydrodynamic conditions (laminar flow)”. In this, it is critical to simulate the condition using solid tablet dosage form in the flow through cell. How did authors simulate to keep consistency for the studied tablets rather than semi solid dosage form?      

Comment 2: In introduction section, I recommend to include various drug related information such as dose, drug instability at different pH, temperature, saturation solubility at varied pH, dissolution number, physicochemical properties of the drug (pka, logP, hydrogen bonding acceptor and donor counts, and aqueous solubility reported). There are various instrumental variation as instrumental error. How did authors keep those constant?     

Comment 3: In material section, authors should write [percent purity of the drug and known impurities. Material section must be expanded for missing other chemicals (solvent and buffer components) used in the study. Moreover, the source of material is missing like city, state, and country. 

Comment 4: In section 2.2.2, the authors withdrawn the sample without replacement. Why is this so? How did the authors maintained the medium volume constant? The UV absorbance value of the drug is 234 nm. Why did the authors set the absorbance value at 273 nm?  The drug was assayed as salt or parent drug? The authors were not familiar with the components of each generic tablet. After the drug release, the composition of the sampled volume was quite different from the fresh release medium. How did authors know the composition of blank set in the UV spectrophotometer?. This causes serious fluctuation in results and assessment of the drug due to interaction. Please justify.

Comment 5: In section 2.2.3, what was the volume for the drug release from the studied tablet? Please reveal the solubility of the drug salt in the studied release medium for each dissolution text. How the temperature 37 C was maintained in the “flow through cell” apparatus?          

Comment 6: What was the detail of standard calibration curve used for the estimation and calculation? Did the authors construct the same in aqueous or organic solvent? Please explain this in a separate section.   

Comment 7: I found various typographical errors in the article. Please revise the manuscript to remove typo and grammatical errors. The language needs to be revised in the revised version of the manuscript.  

Comment 8: Table 1 is not required. Therefore, I recommend to remove from the manuscript. I suggest to combine figure 2 and 3.

Comment 9: The authors have used various abbreviations without explanation. Please expand first appeared in the manuscript text body.  

Comment 10: In the sentence “In this sense, in the present study, the comparison of kinetic parameters (Cmax, AUC0, AUC0, Cmax and Tmax) obtained from non-accumulated dissolution profiles was 623 proposed to establish similarities, as is done for bioequivalence studies, following the 624 FDA and EMA guidelines”, did the authors estimate the PK parameters?. What were the prime reasons for significant differences in dissolution efficiency from one generic product to the other?

Minor correction in the language

Author Response

Dear reviewer ,

The authors are very grateful for the comments and suggestions made to improve the quality of this manuscript before its possible publication. Below are responses to each of the comments made.

Solis-Cruzet al. addressed very interesting findings in their research article entitle as “Discriminative dissolution method using the open-loop 2 configuration of the USP IV apparatus to compare dissolution 3 profiles of metoprolol tartrate immediate-release tablets: use of kinetic parameters”. The work is interesting, well designed, and informative to reader working in the domains. However, I recommend few suggestions before publication.

Comment 1: In the sentence “However, one of the disadvantages of the closed-loop systems of USP â…  and â…¡ apparatuses is that they can mask small differences in API release rates, which can lead to inconsistencies between in vitro and in vivo tests” the authors are recommended to explain the “small differences” generally encountered in the proposed study.  

The sentence was modified to make it clearer, thank you.

Moreover, the authors reported in the sentence as “In this sense, currently, the flow-through cell system (USP IV apparatus) has currently gained more acceptance due to its versatility in dissolution tests, since it is a more discriminatory method due to its hydrodynamic conditions (laminar flow)”. In this, it is critical to simulate the condition using solid tablet dosage form in the flow through cell. How did authors simulate to keep consistency for the studied tablets rather than semi solid dosage form?  

The hydrodynamic conditions which are generated in the USP IV apparatus depend mainly on its engineering and design, as well as the configuration with which the flow cell is used, such as the type of cell, the use of the ruby pearl, the amount and the size of glass beads and even the position of the dosage pharmaceutical form, however these conditions are established during the development of the dissolution test and once established they will be easily reproducible, regardless of the type of dosage pharmaceutical form that is being evaluated.

 Comment 2: In introduction section, I recommend to include various drug related information such as dose, drug instability at different pH, temperature, saturation solubility at varied pH, dissolution number, physicochemical properties of the drug (pka, logP, hydrogen bonding acceptor and donor counts, and aqueous solubility reported). There are various instrumental variation as instrumental error. How did authors keep those constant?  

The equipment and personnel are qualified, as well as the instruments and laboratory materials are calibrated. In addition, the room temperature was kept at 20 ºC and the equipment connected to electric current regulators to avoid voltage fluctuations.

Comment 3: In material section, authors should write [percent purity of the drug and known impurities. Material section must be expanded for missing other chemicals (solvent and buffer components) used in the study. Moreover, the source of material is missing like city, state, and country. 

Missing materials were included in section “2.1. Materials”, as suggested.

Comment 4: In section 2.2.2, the authors withdrawn the sample without replacement. Why is this so? How did the authors maintained the medium volume constant?

In accordance with the provisions of the USP or EP, as long as no more than 10% of the dissolution medium is extracted, sampling can be carried out without replacing the medium. In our case, there were 15 sampling points and in each of these 5 mL of sample was collected, so the inal total volume collected was 75 mL out of 900 mL total. Sink conditions were always maintained since the solubility of metoprolol tartrate is very high (1000 mg/mL) and the tablets evaluated were 100 mg metoprolol tartrate. In this sense, as the volume of dissolution medium varied at each sample point, the adjustment was made when determining the amount dissolved.

The USP IV Flow-through cell is a dissolution apparatus which is described in all international pharmacopoeias, and therefore, before its use for drug product analysis, it needs to be calibrated. In this sense, the dissolution apparatus consists of a reservoir containing the dissolution/release medium, a pump that forces the medium upwards through the vertically positioned flow-cell, and a water bath to control the temperature in the cell. In addition, the equipment has sensors that constantly monitor the temperature of the dissolution medium throughout the test.

The UV absorbance value of the drug is 234 nm. Why did the authors set the absorbance value at 273 nm?  

According to the UV-Vis absorption spectrum of metoprolol tartrate obtained in our laboratory with a solution of approx. 50 mcg/mL using as diluent HCl buffer pH 1.2, acetate buffer pH 4.5 and phosphate buffer pH 6.8, the lengths of maximum absorption were at 225 nm and 273 nm.

Therefore, we decided to work with the length of 273 nm since the intensity in the analytical response was lower to avoid making dilutions in the dissolution samples. That is, the proposed calibration curve was linear in a concentration range of 18.42 to 184.20 mcg/mL (abs: 0.07 to 0.7).

The drug was assayed as salt or parent drug?

The chemical entity used was in the form of a salt (Metoprolol Tartrate) and its solubility is greater than 1000 mcg/mL.

The authors were not familiar with the components of each generic tablet. After the drug release, the composition of the sampled volume was quite different from the fresh release medium. How did authors know the composition of blank set in the UV spectrophotometer? This causes serious fluctuation in results and assessment of the drug due to interaction. Please justify.

It is a fact that we did not know the composition of the matrix (tablet) of the generic drug, but to minimize the effect of the matrix, an analytical method was validated by standard additions, in which the dissolution medium was considered as the adjustment blank. The standard addition method proposed is a somewhat different method from those that can be found in already published articles, since in addition to considering the effect of the excipients, the effect of the amount of excipients is also considered. Now we cannot give more detail as the article is under review.

Comment 5: In section 2.2.3, what was the volume for the drug release from the studied tablet? Please reveal the solubility of the drug salt in the studied release medium for each dissolution text. How the temperature 37 C was maintained in the “flow through cell” apparatus?          

The volume depends on the flow rate of the dissolution medium, which was 8 mL/min, as well as on the sampling times. As the dissolution samples were collected manually every minute for 8 min, the volume sample for each of these points was 8 mL, then every 2 min until reaching 20 min, collecting a sample volume of 16 mL at each of these points, and finally, every 5 min until completing 40 min, with sample volumes of 40 mL for each of these points. As we already described in the introduction section, we evaluated metoprolol tartrate, a salt form of metoprolol which is highly water soluble (>1000mg/mL), so there was no possibility of saturation of the dissolution medium or loss of sink conditions.

Comment 6: What was the detail of standard calibration curve used for the estimation and calculation? Did the authors construct the same in aqueous or organic solvent? Please explain this in a separate section.   

As explained previously, metoprolol tartrate is a chemical entity that is highly soluble in aqueous solvent, therefore, there was no need to use organic solvents, that is, the dissolution medium was always used to prepare the stock solution and the standard solutions. An apology, but we cannot include a section where information related to the analytical method is included since, as we mentioned, there is an article under review that contains this information.

Comment 7: I found various typographical errors in the article. Please revise the manuscript to remove typo and grammatical errors. The language needs to be revised in the revised version of the manuscript.  

Article was reviewed by a native speaker.

Comment 8: Table 1 is not required. Therefore, I recommend to remove from the manuscript. I suggest to combine figure 2 and 3.

Although it may seem that the information in Table 1 is not relevant, the equations used for data analysis in this article may differ from the equations used by other authors, and therefore mathematical deductions may be somewhat different. Also, if we combine Figures 2 and 3, the number of dissolution profiles that the new single figure would show would make it an extremely difficult graph to interpret and it would also be difficult to join since the sampling times are very different, that is, they do not have the same scale.

Comment 9: The authors have used various abbreviations without explanation. Please expand first appeared in the manuscript text body.  

The article was revised, and the meanings of the abbreviations were included.

Comment 10: In the sentence “In this sense, in the present study, the comparison of kinetic parameters (Cmax, AUC0, AUC0, Cmax and Tmax) obtained from non-accumulated dissolution profiles was 623 proposed to establish similarities, as is done for bioequivalence studies, following the 624 FDA and EMA guidelines”, did the authors estimate the PK parameters?. What were the prime reasons for significant differences in dissolution efficiency from one generic product to the other?

Pharmacokinetic parameters were not estimated because an in vivo study was not carried out, but kinetic parameters were estimated, which were calculated from the dissolution profiles and were used to establish similarity between the generic and reference drugs. To establish similarity between the generic and reference drugs, the FDA bioequivalence guidelines studies were followed since in the open-loop configuration non-accumulated dissolution profiles that resemble a pharmacokinetic profile can be obtained.

The main reasons why there were differences in the dissolution efficiency between the drugs is mainly related to the formulation components (excipients) and/or manufacturing processes, since it is evident that the dissolution process is very different and was more evident in the dissolution profiles obtained in the USP IV apparatus.

Reviewer 3 Report

The topic of this manuscript is important and current. The work is well written with the support of current literature. However, changes have to be entered into the revised version of the manuscript before it can be further processed:

1.       USP has a monograph for metoprolol. Why was this dissolution method not included in the presented work?

2.       The innovativeness of research should be emphasized

3.       Please unify: in vitro and in vivo - sometimes italic and sometimes not (page 2 and on); f2 where 2 should be subscripted and f in italic (page 2 and on)

4.       Figure 1 - on the description of the y-axis – “Metoprolo”, and it should be Metoprolol

5.       Table 2 - which values are marked with bold? - please add an explanation

6.       Table 8 - an explanation of the abbreviations MSE, AIC should be added

Author Response

Dear reviewer,

The authors are very grateful for the comments and suggestions made to improve the quality of this manuscript before its possible publication. Below are responses to each of the comments made.

The topic of this manuscript is important and current. The work is well written with the support of current literature. However, changes have to be entered into the revised version of the manuscript before it can be further processed:

  1. USP has a monograph for metoprolol. Why was this dissolution method not included in the presented work?

It is a fact that USP has the metoprolol tartrate tablets monograph, but it is important to mention that a dissolution method is used to evaluate a quality attribute, and a method to compare dissolution profiles must be discriminative enough to be able to determine if a test drug is similar to the reference drug and to some extent ensure bioequivalence. For this reason, the methods for comparing dissolution profiles can be different from those used to assess dissolution (Q) as a quality control test.

  1. The innovativeness of research should be emphasized.

We were reviewing how the emphasis of the research could be improved, but we arrived to the conclusion that what is presented in the manuscript is enough since in the abstract (L.16-22), the introduction (L. 76-81) , the results and discussion (L.365-373, L. 690-765) and the conclusion (L. 801-818) highlight the purpose of the study. However, if you think that innovation should be reinforced or emphasized at some point in the manuscript, it would be a good idea to point it out to us.

  1. Please unify: in vitro and in vivo - sometimes italic and sometimes not (page 2 and on); f2 where 2 should be subscripted and f in italic (page 2 and on)

The manuscript was revised to correct stylistic errors.

  1. Figure 1 - on the description of the y-axis – “Metoprolo”, and it should be Metoprolol

Fixed the bug, thank you.

  1. Table 2 - which values are marked with bold? - please add an explanation

The explanation of the values in bold was included at the foot of the table.

  1. Table 8 - an explanation of the abbreviations MSE, AIC should be added

The explanation of each abbreviation was placed.

Round 2

Reviewer 2 Report

The authors have revised the manuscript as per the suggestions. It can be recommended for publication.

Reviewer 3 Report

Accept in present form